



# Meteorological context of the onset and end of the rainy season in Central Amazonia during the 2014-15 Go-Amazon Experiment

Jose A. Marengo[1], Gilberto F. Fisch[2], Lincoln M Alves[3], Natanael V. Sousa[3], Rong Fu[4],
Yizhou Zhuang[4, 5]

[1]Centro Nacional de Monitoramento e Alerta de Desastres Naturais (CEMADEN), São Jose dos Campos, São Paulo, Brazil
[2] Instituto de Aeronáutica e Espaço/Centro Tecnológico Aeroespacial (IAE/CTA), São Jose dos Campos, São Paulo, Brazil
[3] Centro de Ciência do Sistema Terrestre/Instituto Nacional de Pesquisas Espaciais (CCST INPE), São Jose dos Campos, São Paulo, Brazil
[4] Department of Atmospheric and Oceanic Sciences, University of California, Los Angeles, California, USA
[5] Department of Atmospheric and Oceanic Sciences, School of Physics, Peking University, Beijing, China

*Correspondence to*: Jose A. Marengo (jose.marengo@cemaden.gov.br)

**Abstract.** The onset and demise of the rainy season in Amazonia are assessed in this study using meteorological data from the Go Amazon experiment, with focus is on the 2014-15 rainy season. In addition, global reanalyses are also used to identify changes in circulation leading to the establishment of the rainy season in the region. Our results show that the onset occurred in January 2015, 2-3 pentads later than normal, and the rainy season during austral summer of 2015 exhibited several periods with consecutive dry days in both Manacapuru and Manaus, which are not common for the wet season, and thus determining below normal precipitation. The onset of the rainy season has been strongly associated with changes in large-scale weather conditions in the region due to the effect of the MJO. Regional thermodynamic indices (CAPE, CIN) and the height of the PBL did not present a significant difference between the onset and demise of wet season 2015. This suggests that local changes such the regional thermodynamic characteristics may have not influenced the onset of the rainy season. Variability of the large-scale circulation was responsible for regional convection and rainfall changes in Amazonia during the austral summer of 2014-15.

## 1.    Introduction

The Amazon region represents one of the main convective centers in the world tropics, together with Equatorial Africa and the Indian monsoon regions. In this region, tropical convection is one of the key processes regulating the climate system, and plays an important role in the maintenance of the equilibrium of water and energy balance, and also helping the developing Hadley and Walker circulation. The interannual rainfall variability in the Amazon basin is linked to variations of sea surface temperatures (SST) in both tropical Pacific and Atlantic oceans (Marengo 1992; Ronchail et al. 2002; Yoon and Zeng 2010; Marengo and Espinoza 2015, and references quoted therein). Indeed, previous studies have documented that warm





conditions in the equatorial Pacific (e.g. El Niño events) produce a rainfall deficit in Amazonia, which can originate extreme drought periods, as observed in 1926, 1983, 1997-1998 and 2010 (Williams et al. 2005; Marengo et al. 2008, 2011; Espinoza et al. 2011; Marengo and Espinoza 2015).

In the early 21$^{st}$ large-scale extreme seasonal events, such as extreme droughts in 2005, 2010 and 2015 and floods (2009,
2012, 2014) have affected the Amazon region. Rainfall anomalies were consequence of circulation changes forced by anomalous warming or cooling of the tropical Pacific and/or tropical north or south Atlantic Oceans as documented by Marengo and Espinoza (2015) and references quoted therein). The occurrence of extreme weather and hydro-climate events has change people's perception of climate extremes — as happened after those extensive droughts and flooding. These events classified at the time of their occurrence as "one-in-100 year event", as well as their impacts in natural and human
systems in the region, shows the vulnerability of population an ecosystems in the region to the occurrence of hydro meteorological extremes in the region.

Rainfall in the Amazon basin is mainly supported by the moisture flux from the equatorial Atlantic associated with the trade winds (Angelini et al. 2011). However, Amazon climate have to be seen as coupled interactive atmosphere-ocean-land phenomena (Runyan et al., 2012) in which the land surface plays an equally important part as the ocean, and Makaireva et al.
(2013) have explored the role of the forest in the water recycling, suggesting an active role of the vegetation in the regional water cycle. Previously, Salati and Vose (1984) quantified this influence to be 50 % for inland Atlantic moisture and 50% for local recycling by evapotranspiration and precipitation, using isotopes techniques.  An intensification of the hydrological cycle in Amazonia over the last two decades has been identified by Gloor et al. (2013), caused by an increase in atmospheric water vapour coming from a warmer tropical Atlantic.  This is consistent with a positive trend in precipitation in the
northwestern Amazon since 1990, and this is also reflected in Manaus water levels and in the Amazon discharges at Óbidos (Marengo and Espinoza 2015).

Variability of wet and dry seasons suggests that the onset and demise of the wet and dry seasons and thus the length of the wet and dry season are changing with time (Marengo et al. 2012; Marengo and Espinoza 2015). Observational studies in southern Amazonia suggest that the dry season has increased in length by about one month since the 70's (Marengo et al.
2011; Fu et al. 2013).  Furthermore, the length of the dry season also exhibits interannual and decadal-scale variations linked either to natural climate variability, or as suggested by Wang *et al.* (2011) due to the results of the influence of land use change in land use in the region.   While it is important to know how will be the characteristics of the rainy season total rainfall, it is also important to highlight the urgency for improving our understanding and capability to detect and predict the rainy season onset and demise, as well as the wet and dry seasons variability, not just through model experiments but also
through observational analysis. The droughts of 2005 and 2010 and their impacts of humans and on the tropical forest have been characterized by late onsets of the rainy season and/or longer dry seasons (Marengo et al. 2011; Marengo and Espinoza 2015). During the recent El Nino in 2015-16, rainfall over central-northern Amazonia has been below normal (approximately 200-300 mm below normal from wet season), and this contributed to an extensive drought and subsequent problems in the



hydrology of the region, as well as an increased the number of fires in the region (CPTEC-www.cptec.inpe.br). The length of the dry season has a strong temporal and spatial variability, and is strongly associated with dry conditions that may impact the occurrence of fires and their impacts on the release of carbon and aerosols, and also affecting human systems (Aragao et al. 2016; Martin et al. 2016).

Various studies have discussed observational and model aspects of the onset and demise of the rainy season in Amazonia using a variety of climatic indicators, such as rainfall, outgoing long wave radiation or dynamic fields (e.g., Kousky 1988; Marengo et al. 2001, 2012; Fu et al. 1999; Liebmann and Marengo 2001; Gan et al. 2004; Wang and Fu 2004; Silva et al. 2007; Silva and Carvalho 2007; Raia and Cavalcanti 2008; Silva 2009; Marengo and Espinoza, 2015). However, modelling work still show uncertainties in the representation of the onset of the rainy season. This may be due to the poor

representation of clouds and land surface-atmosphere interactions or due to role of aerosols and other particles, which are still not well represented in models.

Moisture transport across the equator and its variations could influence convection and thus the wet season onset (Rao et al. 1996; Marengo et al. 2001; Wang and Fu 2002; Alves 2016).  Li and Fu (2006) showed that weak and infrequent extratropical cold front penetrations during the transition season also contribute to a delay of the wet season onset.  However,

the complexity of the relationship between ENSO, Atlantic SSTs and the wet season onset over the southern and central Amazon remain unclear. Butt et al. (2011) identify significant differences on the onset of the rainy season in Rondonia between 1970 and 2000, due to land use changes in the region. However the uncertainties are still high on the attribution of these extremes and their variations to natural and human influences. This highlights the importance and urgency of understanding the underlying causes of the onset and demise of the rainy season and our ability to predict them.

Furthermore, evidences on the possible role of human influences (deforestation, increase of greenhouse gases and aerosol released due to biomass burning or urban pollution) on rainfall and river variability have started to appear in the literature recently (Cecchini et al. 2016; Alves 2016; Sparcklen and Garcia-Carreras 2015; Magrin et al. 2014; Zhang et al. 2008).

Increasing aerosol concentrations can have substantial impacts on spatial and temporal rainfall patterns in the Amazon (e.g., Martins et al. 2009a; Reutter et al. 2009; Pöhlker et al. 2016). The initial work by Andreae et al. (2004) suggest that aerosols

from biomass burning in Amazonia may delay the onset of the rainy season in southern Amazonia, but not much is known on the possible roles of aerosols from urban areas in rainfall and the water cycle in the Amazon region.  Previous studies have identified that cloud microphysical properties, cloud cover, precipitation, lightning, and regional climate over the Amazon basin can be significantly affected by aerosol particles (Lin et al. 2006; Rosenfeld et al. 2008, 2014; Martins and Silva Dias 2009; Altaratz et al. 2010; Koren et al. 2012; Gonçalves et al. 2015; Wang et al. 2016).

In this paper, we use data from Go Amazon and CHUVA experiments during 2014-15 as well as global reanalyses to investigate regional and large-scale circulation and rainfall patterns during the onset and demise of the rainy season. Emphasis is on the identification of large-scale patterns leading to the onset and demise of the rainy season in the Manaus region in central Brazilian Amazonia in both years. We have taken the advantage of the high resolution of surface meteorology data collected during these two field experiments part of Go Amazon in 2014-15 and also the presence of El





Nino during summer of 2015, to investigate daily and diurnal rainfall variability. We also investigated the large scale and regional circulation patterns linked to rainfall variability in those regions. With the Go Amazon data we have investigated the onset and demise of the rainy season of 2014-15, as well as convection and the planetary boundary layer (PBL) heights in some sites near Manaus, where thermodynamic indices were calculated to identify the transition regimes pre and post onset

and demise of the rainy season.

## 2.      Methods

The data used for this study comes from the Go Amazon Project (Martin et al. 2016), designed to study some of the characteristics on the rainy season in Amazonia, such as onset and demise of dry and wet seasons and we have used some of

the CHUVA and Go Amazon 2014/15 rainfall and surface and upper air meteorology data available from 2014 to 2015. The expression green ocean (Go) was introduced by Williams et al. (2002) due to the similarities in aerosol particle concentrations and cloud microphysics between the Amazon basin and remote oceanic regions during clean periods of the wet season. Observations and Modeling work of the Go Amazon Experiment (Go Amazon 2014/15-(http://campaign.arm.gov/goamazon2014/) were collected in the central region of Amazonia near Manaus from 1 January

2014 through 31 December 2015. More details on the nature and objectives of Go Amazon 2014/15 can be found in Martin et al. (2016).

As described by Machado et al. (2014), the CHUVA project—CHUVA, meaning "rain" in Portuguese, is the acronym for the Cloud Processes of the Main Precipitation Systems in Brazil: A Contribution to Cloud-Resolving Modeling and to the Global Precipitation Measurement (GPM). It began in 2010 and has conducted five field campaigns while the last

experiment was held in Manaus as part of the Go Amazon 2014/15 experiment. The CHUVA's main scientific motivation is to contribute to the understanding of cloud processes, which represent one of the least understood components of the climate system. Field data from the CHUVA and Go Amazon 2014-15 campaigns are used to improve our understanding the dynamics of the onset of the rainy season and the characteristics of the dry season on that year comes handily, since we have the possibility to identify the onset the rainy season over central Amazon using high resolution meteorological data collected

nearby Manaus since the end of 2013. In addition, we analyse ground-based remotely sensed data (ceilometer) to identify changes in the planetary boundary layer (PBL) during the onset and demise of 2014-2015 rainy season. Wind, Outgoing Longwave Radiation (OLR) and Sea Surface Temperature data from National Center for Environmental Prediction (NCEP) / Climate Prediction Center (CPC) are used to examine the influence of atmospheric and oceanic conditions on the onset and demise of the rainy seasons.

Rainfall data (at diurnal and daily level) comes from the Brazilian Meteorological Service (INMET) station at Manaus (Lat. 3.11$^{o}$S – Lon. 59.95$^{o}$W) and at Manacapuru from State of University of Amazonas UEA (70 km upwind of Manaus Lat.: 03.05$^{o}$S – Lon. 60.00$^{o}$W). The pentad of the onset and end (or demise) of the rainy season in Amazonia was calculated using the criteria of rainfall accumulation data defined by Liebmann and Marengo (2001) and adapted by Bombardi and Carvalho (2009) using gridded rainfall data. This criterion was applied to data from the Manaus and Manacapuru (nearby Manaus)





rainfall stations from the Go Amazon-CHUVA network during 2014-15, by averaging all available data for a given day from those two stations onto a 1.0° grid. The availability of data from a large number of stations in a grid box as allows a more regional focus as compared with single station data.

Additional data sets for regional rainfall analyses during those two rainy seasons come from the CPC/NCEP/NOAA
(www.ncep.noaa.gov) and from the Global Precipitation Climatology Project (GPCC) gauge-based gridded precipitation dataset, available for the global land surface only (Rudolf et al. 1994). The GPCC datasets are available in the spatial resolutions of 1.0° latitude/1.0° longitude as mean monthly precipitation totals and anomalies from the long-term mean for 1951–2000

Once the pentad of the onset was identified for Manaus and Manacapuru, various analyses are performed before and after the
onset, in order to identify and understand possible shifts in atmospheric circulation and energy fluxes and characteristics of the PBL and thermodynamics indices Convective Available Potential Energy (CAPE) and Convective INhibition energy (CIN) that would favour the establishment of the onset of the rainy season of 2014-15. CAPE and CIN values were computed using the original variables for the 500 m mixing layer parcel. The height of the PBL was derived from a ceilometer installed at the T3 site in Manacapuru and its hourly values were computed for pentads pre and post onset and
demise of the rainy season in 2014-15. The Latent (LE), sensible (H) turbulent heat fluxes and the Bowen ratio derived from these fluxes at the T3 site and the EMBRAPA Flux for 2014-15 in order to quantify their values before, during and after the onset and end (or demise) of the rainy season, so we can investigate the energy partition and convective processes that accompany the evolution of the rainy season during the days of the Go Amazon campaign. More details about the instrumentation used can be found in Machado et al. (2004), Martin et al. (2016) and Wang et al. (2016).

## 3.    Results

### 3.1    Characteristics of the 2014-15 rainy season in Amazonia

The mean climatic features of the Manaus region are described elsewhere (Greco et al. 1990; Cohen et al. 1995; Machado et al. 2004; Martin et al. 2016), where the peak of the rainy season occurs around March-May. The GPCC rainfall (Figure 1a-d)
shows rainfall anomalies from December 2014-February 2015 (representing the wet season) top September-November 2015. Over the central and eastern Amazonia for the wet season rainfall was about 80-90 mm/month below normal, while over western Amazonia rainfall was about 50-90 mm/month above normal. For the SON period the rainfall was well below normal (around 90 mm/month below normal), in almost all Amazonia. This is a signal of impacts of El Nino 2015-16 that was under development in the tropical Pacific since the middle of 2015 (Figure 2). Warm surface waters (1.5-2.5 °C) are
detected along the equatorial Pacific in the period from March to May 2015. This 2015-16 drought caused the longest fire season of the 21st century with five months exceeding 10,000 fire detections and the largest number of active fire occurrences per km² deforested (Aragão et al., 2016, CPTEC, 2015). This combination of a longer dry season, more frequent extreme droughts and an increased risk of fire could play a critical role in a future Amazon rainforest dieback in spite of the



increased resilience of tropical forests in an elevated atmospheric $CO_2$ environment (Huntingford et al. 2013).

On the regional scale circulation features, during DJF2015 it did not show signals of El Niño in the tropical Pacific while the warm surface waters are already present during MAM 2015 (Figure 2). This warming increased continuously until March 2016, indicating the intensification of El Niño in 2015-16, with warm surface water (3-4 ºC above normal) over the

equatorial eastern Pacific by austral summer of 2016 (www.cptec.inpe.br). The low level circulation over the tropical North-Atlantic and Amazon sectors (Figure 2) showed reduction in the Northeast trades, suggesting reduced moisture transport from the tropical North Atlantic into the Amazon region in austral summer and fall of 2015. An analysis of the near-surface and upper-air circulation discussed previously can provide a better idea on the regional east-west circulation of the region during December 2014 and January 2015. The east–west circulation along the equatorial zone (5°N–5°S) in Figure 3 shows

upward motion anomalies over western Amazonia during both summer months while reduced convection and downward motion with subsidence is found over Central Amazonia, Eastern Amazonia and Northeast Brazil. The later region is experiencing a record drought since 2011 (Marengo et al. 2016). These circulation anomalies are consistent with negative rainfall anomalies over Central Amazonia nearby the Manaus region. Therefore, interannual variations of the wet season onset in the Amazon appear to be influenced by changes in large scale and regional circulation over the tropical and Pacific

sectors.

3.2     Daily rainfall variation and characterization of the onset and demise of the rainy season during 2014-15.

Following Liebmann and Marengo (2001), the CPC/NCEP/NOAA data showed that the climatological onset of the rainy season for the Manaus region is detected around the pentad 70 (12-16/December) and the demise of the rainy season occurs

around pentad 32 (05-09/June). For the wet season 2014-2015 there was a delay of the rainy season, with the onset occurring in pentad 6 (26-30/January 2015) and the demise occurred at pentad 26 (06-10/May 2015), meaning a rainy season shorter than normal. Figure 4 shows the details of the onset and demise of the rainy season near Manaus for the climatology and during 2014-15, noticing that the onset of the rainy season in 2015 occurred 2-3 pentads later than normal. On the other hand, during the summer of 2014-15 there were several consecutive dry days between December 2014 and March 2015 in

both Manacapuru and Manaus (Figure 4), which are not common for the wet season.

As seen in the previous section, despite the oceanic and atmospheric conditions in the equatorial Pacific (Niño 1 + 2 and 3) which show ENSO-Neutral conditions during the period before the onset it is noted that the patterns of regional precipitation distribution over the Central and Eastern Amazonia were consistent with that expected for a ENSO event (Figures 1 and 3). Although this clear influence of the large-scale circulation modes, it is observed that the onset of the rainy season has been

strongly associated with distinct phenomena that caused changes in weather conditions in the region, for example, the Madden-Julian Oscillation (MJO) (Madden and Julian 1994, Liebmann et al. 1999; De Souza and Ambrizzi 2006; Alvarez et





al. 2015). Figure 5 shows the longitude versus time diagram of OLR anomalies between 5ºN-5ºS over the globe in 2014 and 2015.

It is observed that from July through later October 2014 the intraseasonal signal became less coherent, with a weaker anomaly field and it is inconsistent with a canonical El Niño signal. The pattern became more organized during late

November as the MJO strengthened as indicated by eastward propagation of alternating anomalies into January 2015. At this time, the MJO may have contributed to enhance rainfall and the onset of rainy season for portions of Amazonia as indicated by negative OLR anomalies (blue shading) favouring conditions for precipitation around 60ºW.

This fact was reflected in the temporal distribution of rainfall (Figure 4), which notes a regional frequency of precipitation in this period. In summary, the phases of MJO associated OLR anomalies were evident throughout the Equatorial region, and in

particular over the Central and Eastern Amazon. This suggests that the negative MJO phase in mid-January may have contributed to favouring conditions to enable the convection and the onset of rainy season from pentad 6.  This is consistent with other atmospheric mechanisms on local scale, for example, the local circulation and thermodynamic patterns near surface.

Figure 6 shows the daily variation of air temperature at 850 hPa, relative humidity and Bowen ratio at the T3 Manacapuru

site. Wet conditions were consistent with larger relative humidity and lower temperatures, while dry spells occur with lower relative humidity and higher air temperatures. Before the onset we noticed a reduction in temperature and humidity while it is hard to see any tendency of the Bowen ratio before the onset or after the demise of the rainy season.  The latent heat fluxes over the land surface are important sources of atmospheric humidity during the initial stages of the transition season between dry and wet periods (Fu and Li 2004). Together with changes in the onset of the rainy season, changes of dry season

length may be key in favouring present risk of fire.  High land surface Bowen ratio during the preceding dry season would delay the subsequent wet season onset in the southern Amazon (Fu and Li, 2004), and this may have been the case in the onset rainfall in 2014-15.

### 3.3 Thermodynamic indicators and PBL behaviour during the onset and demise of the rainy season in 2014-15

CAPE and CIN have been calculated at daily scale for before and after the onset of the rainy season using the T3 radiosonde data (Figure 7). The CAPE and CIN were calculated for each profile and averaged for a day.  Figure 7 shows that CAPE and CIN are very noisy.  A two-sample t-test for daily average CAPE/CIN values during the rainy season (2015/1/26-2015/5/10) and before/after rainy season (2014/11/1-2015/1/25 and 2015/5/11-2015/12/1) was performed. Results suggest both CAPE and CIN value have significant differences between the two periods (during rainy season and before/after rainy season),

indicating that there is significant change (at 5% significance level) of CAPE and CIN between the days with and without deep convection.



The diurnal cycle of the heights of PBL was computed with the ceilometer installed at T3 site. The Figure 8 shows the composite PBL diurnal cycle for the wet season 2014-2015. The PBL height remained stationary around 300-400 m at night-time (nocturnal boundary layer), then increased during daytime (convective boundary layer), and reached its maximum (1100 – 1200 m) at early afternoon (14:00 LT). This pattern is consistent with the previous values obtained for Amazonia for

wet season (Fisch et al., 2004) and with the thermodynamic indices. There is no signal of the anomalous wet season in the PBL heights. Individual diurnal cycles for each month (Figures 8b,c) also did not present a significant difference between the onset and demise of wet season 2015.

A moistening of the planetary boundary layer and a lowering of the temperature at its top may reduce CIN and controls the conditioning of the large-scale thermodynamics prior to onset (Fu et al 1999). In addition, Li and Fu (2004) found that the

main increase in CAPE and reduction in CIN occur prior to the rainy season onset, although in the tropical atmosphere decrease CAPE often exists in the absence of deep convection (Williams and Renno 1993).

### 4.        Conclusions

The onset of rainy season in Amazonia is assessed in this study based on changes in precipitation, large-scale synoptic flow

fields and thermodynamic parameters during the Go Amazon experiment. The onset and demise of rainy season in Amazonia have been assessed on this study based on changes in precipitation, large-scale synoptic flow fields and thermodynamic parameters during the Go Amazon experiment.

Focus has been on the 2014-15 rainy seasons using the available climatic data from the Go Amazon experiment as well as from other sources. From our results, based on the analysis of daily data from various sites of the Go Amazon field

experiment, it was observed that the wet season of 2014-2015 has a delay of the onset of the rainy season, with the onset occurring in January 2015, 2-3 pentads later than normal. On the other hand, during the rainy season of summer 2015 there were several consecutive dry days between December 2014 and March 2015 in both Manacapuru and Manaus, which are not common for the wet season, and thus determining below normal precipitation. The onset of the rainy season has been strongly associated with changes in large-scale weather conditions in the region due to the effect of the MJO.

The CAPE and CIN do not show any significant change between the days with and without deep convection and while there is an increase in CAPE before the onset and decrease after the demise, no clear change of CIN during onset period is detected. The diurnal cycle of the heights of PBL also does not show any signal of the anomalous wet season and the individual PBL diurnal cycles did not present a significant difference between the onset and demise of wet season 2015. While one of the main objectives of the Go Amazon Experiment was to assess the influence of the air pollution from the city

of Manaus in the rainy season on that region, we do not have evidence to suggest that local changes such the regional thermodynamic characteristics or even the aerosol release from the city of Manaus may have not influenced the onset of the rainy season. Variability of the large-scale circulation was responsible for regional convection and rainfall changes in Amazonia during the austral summer of 2014-15.





**Acknowledgements**

Data were obtained from the Atmospheric Radiation Measurement (ARM) Program sponsored by the U.S. Department of Energy, Office of Science, Office of Biological and Environmental Research, Climate and Environmental Sciences Division. This work was supported by FAPESP/DOE/FAPEAM Go Amazon grant 2013/50538-7.

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





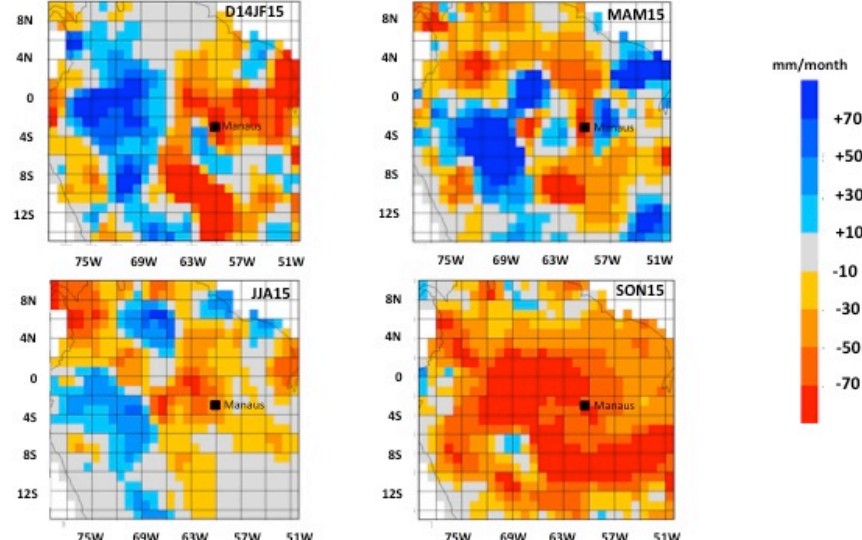

**Figure 1.** **Rainfall anomaly (mm month⁻¹) during December 2014-May 2015 to September-November 2015. Data comes from GPCC and anomalies are relative to the 1951-2001 climatology. Black square indicates the location of Manaus.**



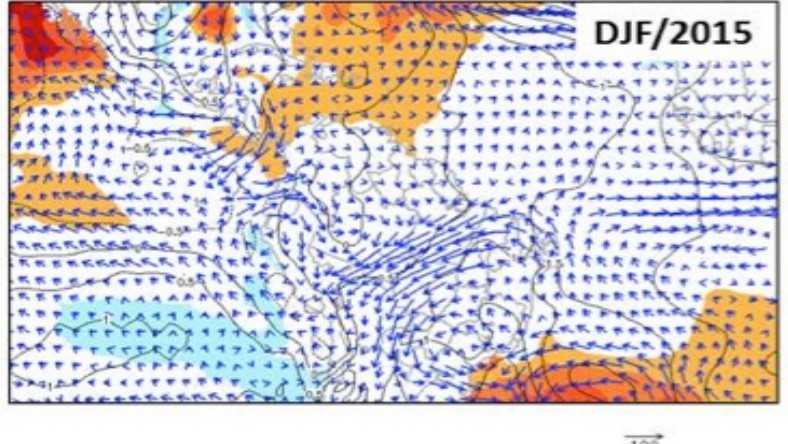

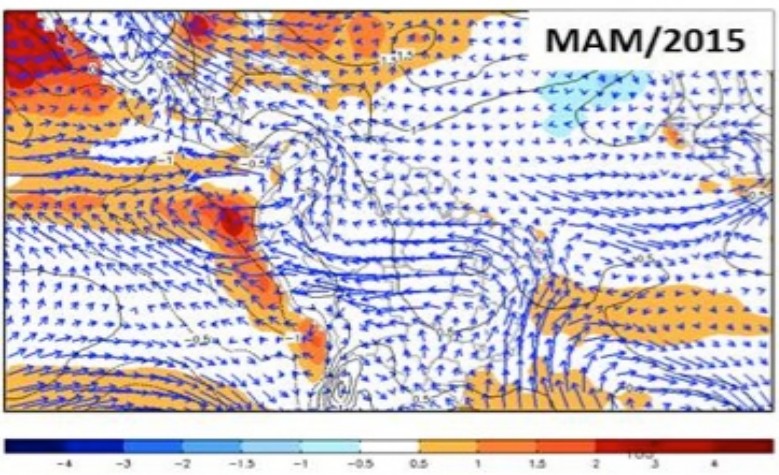

**Figure 2.** Seasonal SST anomalies (ºC), SLP anomalies (hPa), and anomalies of vertically integrated moisture transport (vectors) from the surface to 500 hPa in South America from December 2014 to May 2015. SST and circulation anomalies correspond to the 1961–2012 long-term mean. The bar at the bottom of the panel shows the scale of the SST anomalies. The vector at the bottom of the panel shows the scale of the moisture transport ($kg^{-1}ms^{-1}$). Black full lines show SLP anomalies (hPa)



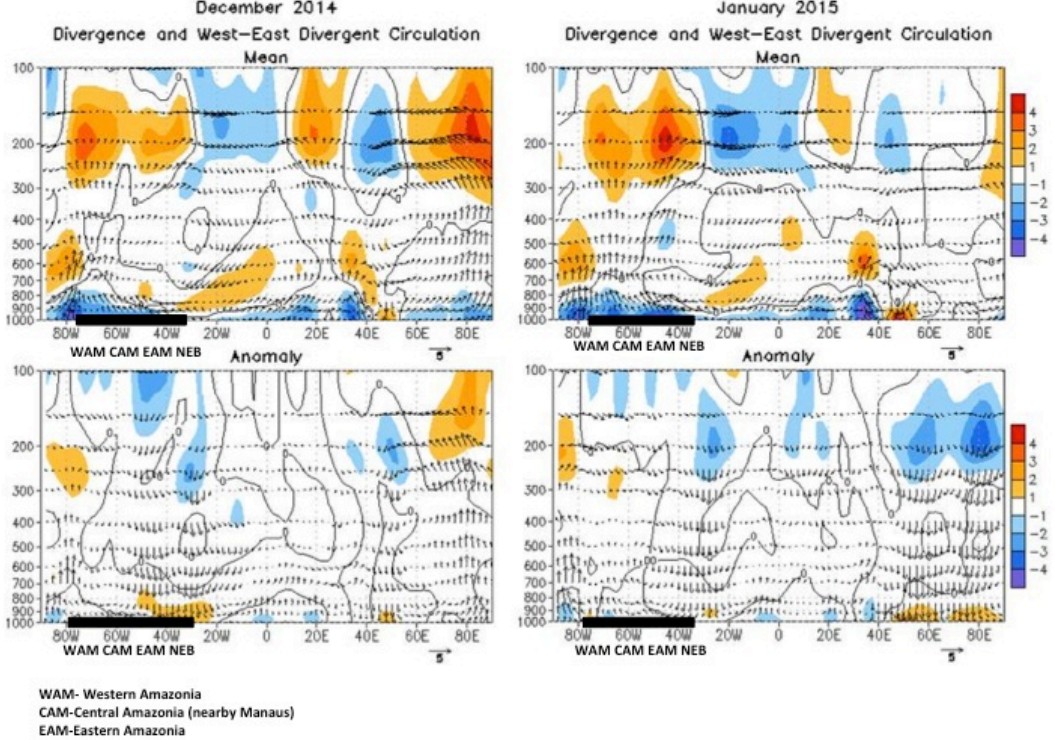

WAM- Western Amazonia
CAM-Central Amazonia (nearby Manaus)
EAM-Eastern Amazonia
NEB- Northe4ast Brazil

**Figure 3. Pressure-longitude section (80˚W-100˚E) of the mean (top) and anomalies (bottom) divergence (counter interval is 1 x 10⁻⁶ s⁻¹) and divergence circulation between 5˚N-5˚S. Vectors of combined vertical velocity and the divergent component of the zonally wind represent the divergent circulation. Red shading and solid contours denote divergence (top) and anomalous divergence (bottom). Blue shading and dashed denote convergence (top) and anomalous convergence (bottom). Anomalies are departures from the 1981-2010 long term mean.**



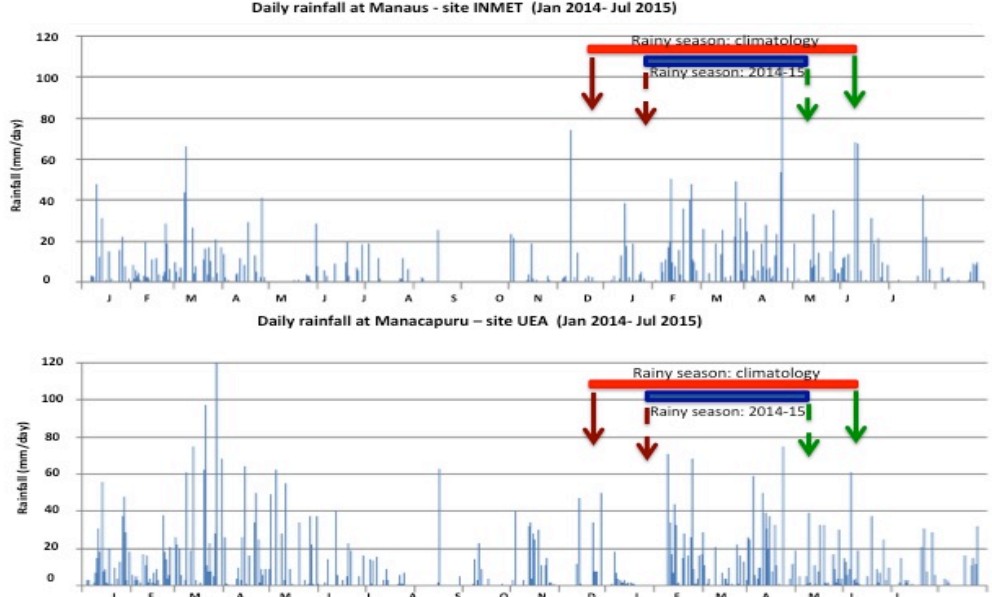

**Figure 4. Daily variation of rainfall (mm day$^{-1}$) at Manaus (INMET site) and Manacapuru (UEA site) from January 2014 to July 2015. Red bar shows the climatological occurrence of the onset and end of the rainy season, while blue bar indicates the onset and end for 2014-15.**





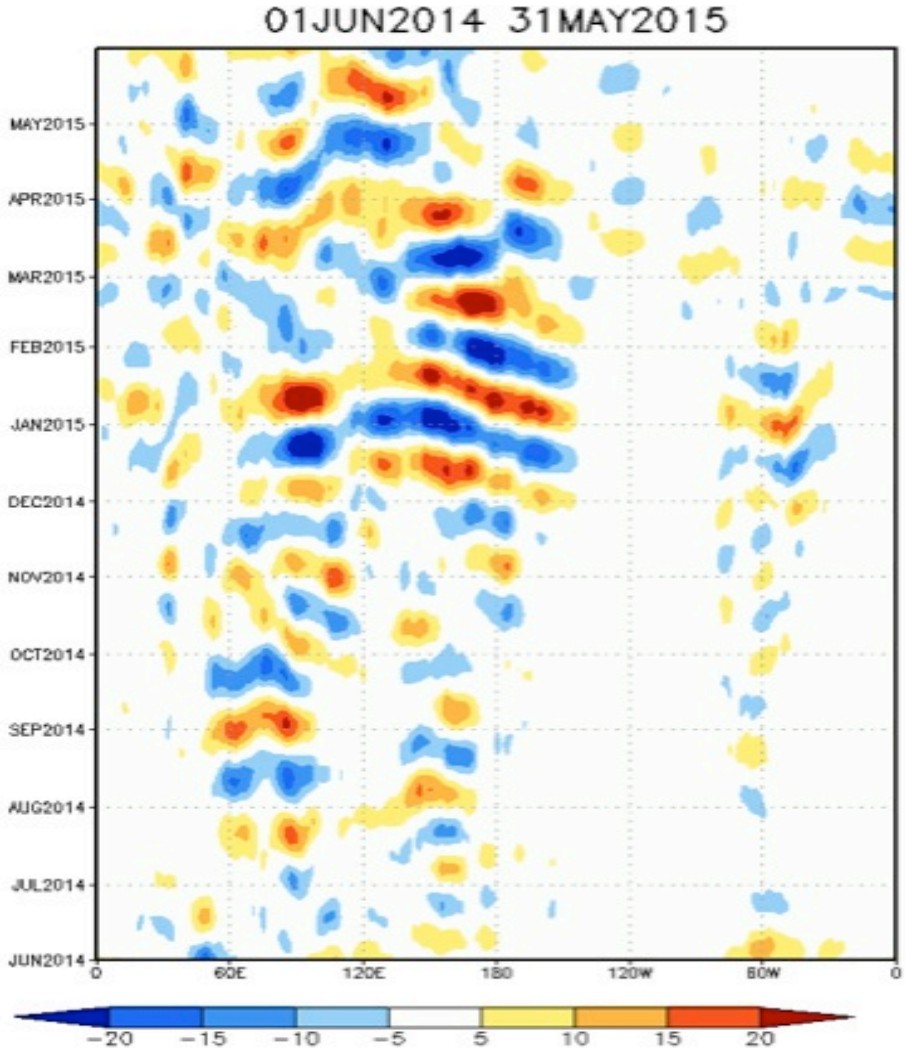

**Figure 5. Hovmoller diagram of 30-60-day filtered OLR anomalies (W m$^{-2}$) along the Equator (5ºN-5ºS) during the**
5  **period of June 2014 to May 2015.**





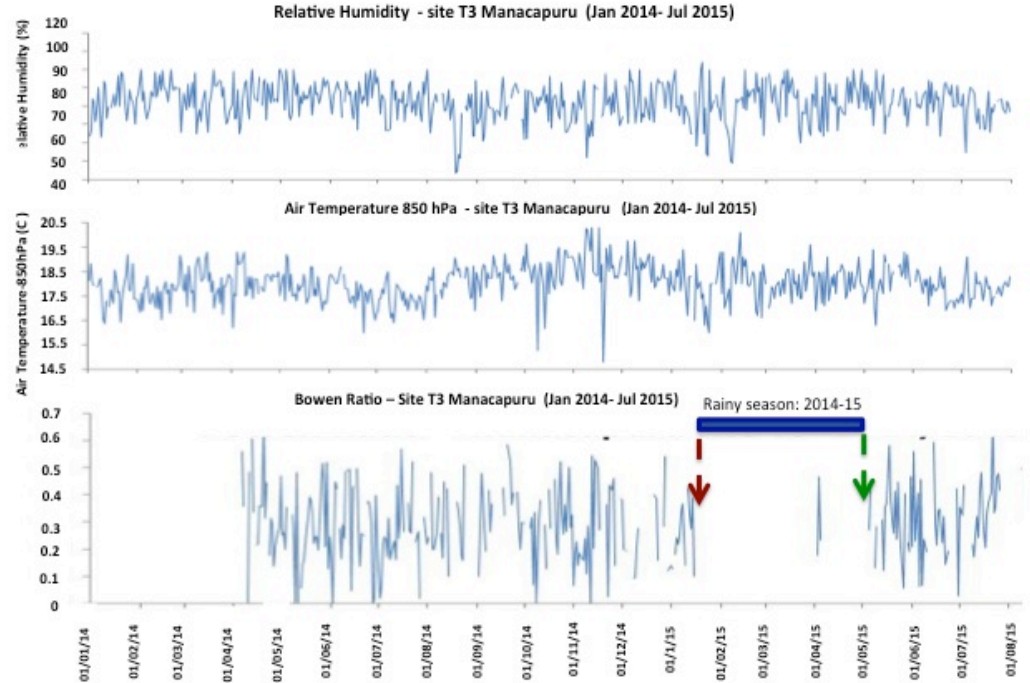

Figure 6. Daily variation of relative humidity (%), 850 hPa air temperature (°C) and Bowen ratio at the T3 Manacapuru site, from January 2014 to July 2015. Blue bar indicates the onset and end of the rainy season for 2014-15.



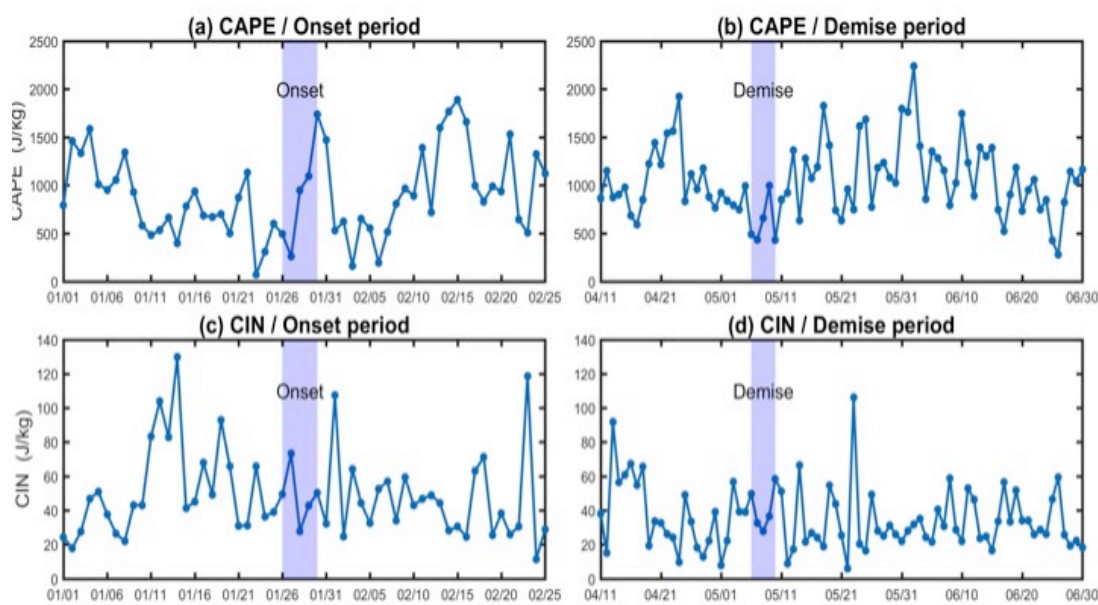

**Figure 7. Daily variation of CAPE and CIN (J kg⁻¹) from January 1 to February 26, 2015 (around the onset of the rainy season) and from April 11 to June 30 (around the demise of the rainy season) at Manaus. Pentads of the onset and demise are identified with a blue vertical line.**




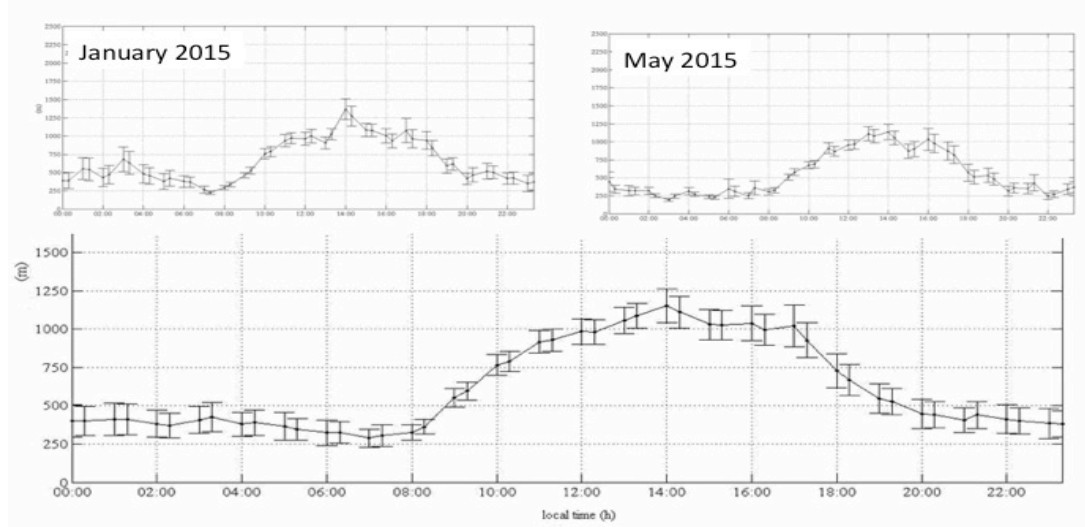

**Figure 8. Diurnal cycle of the height of boundary (m) layer (average from January 1 up to June 30, 2015) (a); from January 2015 (b) and June 2015 (c).**

