# Peer review of "Meteorological context of the onset and end of the rainy season in Central Amazonia during the 2014-15 GoAmazon Experiment"

_Atmospheric Chemistry and Physics, 2017_

## Referee Comment (RC1) · Anonymous Referee #1 · 21 Feb 2017

This should be accepted (with some mandatory revision) because it was written by the great José Marengo.

The comments that I would require to be answered before acceptance are the following:

Page 1, Line 31: "...and also helping the developing Hadley and Walker circulation." Please either correct or elaborate on this phrase. Why is the Hadley circulation developing? Do you mean within the context of the seasonal cycle? And how does Amazon convection participate in the Walker circulation? My understanding is that the Walker circulation is an Indo-Pacific phenomenon.

P1, L35: Can you support your statement that Atlantic SSTs affect Amazon precipitation with a sentence or two, similar to your description of the Pacific influence? And I believe it would be of value to describe, again briefly, the nature of the teleconnections in both the Atlantic and Pacific. For example, I understand the Pacific teleconnection works by shifting the Walker Circulation to the east, so during El Nino there is more subsidence over the Amazon. Since I have no idea how the Atlantic influences the Amazon, it would be nice to have an idea. Again, nothing new here, just remind the reader what other studies have concluded with a physical explanation. It is nice to have a picture of what is going on.

2-8: How have people's perceptions been changed? Please explain a little better with some detail.

2-10: I think it should be the plural, "show" but it is a complicated sentence.

2-13: "has"

2-22: "Variability" sentence does not actually make sense. "Variability" suggests variation, not necessarily long-term change.

2-24 "70s"

2-27: "While it is important to know how will be" - badly written

2-29: "season"

2-33: Explain what you mean by, "problems in the hydrology of the region."

3-9: This sentence repeats itself: "This may be due to the poor representation of clouds and land surface-atmosphere interactions or due to role of aerosols and other particles, which are still not well represented in models."

3-14: What do you mean, " Li and Fu (2006) showed that weak and infrequent extratropical cold front penetrations during the transition season also contribute to a delay of the wet season onset?" I presume you mean weaker and less frequent than usual, but if you do, you need to be specific.

4-2: What? "On the regional scale circulation features, during DJF2015 it did not show signals of..."

4-34: Liebmann and Marengo used gridded rain data.

I give up on the writing. Suffice it to say that it is badly written and needs improvement.

Please use scaleable, or "vector" graphics. Your rasterized graphics appear fuzzy and thus unprofessional.

Figure 2 is not of acceptable quality. In addition to printing it using scaleable graphics, it needs latitudes and longitudes, continental outlines one can see, and perhaps fewer vectors.

6-2: horribly written sentence: " On the regional scale circulation features, during DJF2015 it did not show signals of El Niño in the tropical Pacific while the warm surface waters are already present during MAM 2015..."

6-6: From what do you infer reduced northeasterly trades? Is it from the vectors, even though the quantity present by the vectors is the integral to 500 hpa, or is it from assuming flow nearly parallel to surface contours?

6-5: Assuming that you are using the vectors to make the statement, "The low level circulation over the tropical North- Atlantic and Amazon sectors (Figure 2) showed reduction in the Northeast trades....," I disagree (assuming my guess about the map domain is correct). Yes, along the equator (assuming Fig. 2 is centered on the Equator), there are westerly anomalies, but these are away from the coast (looking at DJF). Along the Atlantic coast and north of the equator, however, the anomalies are near-zero. There are huge positive transport anomalies from the equator into the southern Amazon, which are consistent with above-normal precipitation to the west (south of the equator), as there appears to be anomalous convergence of moisture there (Fig. 2a). So, please explain why this is inaccurate and why your statement is correct.

6-7: Why are Figures 1 and 2 made from seasonal averages, while Fig. 3 is from

monthlies? Would it not be better and certainly more consistent to use seasonal averages in Fig. 3?

6-10: The authors may have a valid point, but I believe they should hone in more on Brazil. Nothing is discussed east of the GM, so why not just show the longitudes of South America, plus or minus a bit? And perhaps the shading interval on the anomalous maps should be lowered, as with the present interval it doesn't look like much is going on over South America.

6-13: " Therefore, interannual variations of the wet season onset in the Amazon appear to be influenced by changes in large scale and regional circulation over the tropical and Pacific sectors."

6-21: Instead of, "meaning a rainy season shorter than normal" how about, "meaning a rainy season that was shortened at both ends."

Figure 4 is a little disturbing to me because it does not appear the INMET and UEA records match very well the NOAA records. Looking at the bottom record (Manacapuru), there is no rainfall at all within several days of onset, and it continues to rain for at least a week or so after the NOAA end. I know Manaus is a long record and I assume so is Manacapuru, so why not use the daily station data to do the onset and end calculations? You know that the actual station data is the best record available, so I don't see any reason to use NOAA. I think your point could be made more succinctly and more accurately.

6-25: "which are not common for the wet season." This cannot be stated without any sort of justification, such as a reference.

Conclusions: Please make sure your conclusions match your discussion in the Results section. For example, you discussed the change in moisture transport (which I disagreed with), but this discussion did not make it into the conclusions.

Good luck - Brant Liebmann

---

## Short Comment (SC1) · 22 Feb 2017

This should be accepted (with little revision) because it is a very interesting theme. This research has great impact, since the determination of the dry and rainy season of the Amazon is important not only locally but for the distribution of humidity in the rest of the South American continent. Studies with high frequency data are highly desirable as they describe in detail the characteristics of the region. In this context, this work is able to encompass the local characteristics of the region, with data from the experiments Go Amazon and Rain Project and the large scale characteristics due to the ENSO phenomenon. In this study the authors suggest that the dry season experienced has impacts of the En Nino and the beginning of the rainy season may be related to the

MJO.

- How does the relationship between MJO and ENSO occur? A brief explanation of this relationship would contribute to a better understanding of the article. - It is known that different phases of the MJO influence the precipitation in South America in different seasons. What is the climatological influence of MJO in this region during the year, and how is it different from that found in this specific study?

---

## Short Comment (SC2) · 1 Mar 2017

S. Ferraz simonetfe@gmail.com

This should be accepted (with little revision) because it is a very interesting theme. This research has great impact, since the determination of the dry and rainy season of the Amazon is important not only locally but for the distribution of humidity in the rest of the
South American continent. Studies with high frequency data are highly desirable as they describe in detail the characteristics of the region. In this context, this work is able to encompass the local characteristics of the region, with data from the experiments Go Amazon and Rain Project and the large scale characteristics due to the ENSO phenomenon. In this study the authors suggest that the dry season experienced has impacts of the En Nino and the beginning of the rainy season may be related to the MJO. - How does the relationship between MJO and ENSO occur? A brief explanation of this relationship would contribute to a better understanding of the article. - It is known that different phases of the MJO influence the precipitation in South America in different seasons. What is the climatological influence of MJO in this region during the year, and how is it different from that found in this specific study?

Response:

Broadly speaking, the influence of the MJO on precipitation over the tropics occurs by eastward propagation of Rossby wave trains from the tropical Pacific Ocean (Muza et al. 2009). Previous observational and modeling studies generally indicated that MJO and ENSO has a decadal variation and seasonal dependence (Tang and Yu, 2008; Hendon et al. 2007), however, them has not been well identified due to nonlinear in nature. These studies also show significantly lagged correlations between MJO and ENSO indices.

Despite this, recently, Shimizu et al. (2016) examined the regional relationship between ENSO and MJO phases on climatological patterns of precipitation over South America. The results indicated that combined responses showed that precipitation is strongly influenced by the MJO phases rather than by ENSO conditions, especially during the austral summer. Then, our results corroborate with Shimizu et al. (2016) who observed highest percentages of days with active MJO occurred during El Niño and neutral years and an increase of precipitation.

References:

Hendon, H. H., M. Wheeler, and C. Zhang (2007), Seasonal dependence of the MJO-ENSO Relationship, J. Clim., 20, 531 – 543.

Tang, Y., and B. Yu (2008), MJO and its relationship to ENSO, J. Geophys. Res., 113, D14106, doi:10.1029/2007JD009230.

Muza MN, Carvalho LMV, Jones C, Liebmann B (2009) Intraseasonal and interannual variability of extreme dry and wet events over Southeastern South America and Subtropical Atlantic during the Austral Summer. J Clim 22:1682–1699.

Shimizu, M.H. & Ambrizzi, T. Theor Appl Climatol (2016) 124: 291. doi:10.1007/s00704-015-1421-2. –

Please also note the supplement to this comment:
http://www.atmos-chem-phys-discuss.net/acp-2017-22/acp-2017-22-SC2-supplement.pdf

---

## Author Response (AR2)

New reviews by co editor Silva Dias:

Please incorporate the reviewers suggestions.

Non-public comments to the Author:
The responses to the reviewer´s comments have not been fully incorporated in the text. The reviewers have indicated important points that required further explanation not only to the reviewers but to others that may eventually read this article.

**Response:  We have included all comments in response to reviewers  in the text of the paper.**

The manuscript still has many grammatical errors that are expected to be corrected for the final version.

**Response:  The text has been revised by a US native speaker that we hired.**

Page 7 lines 25-30. the student t-test is applied, what is the significant change found in CAPE and CIN? Is this consistent with the sentence in line 25 page 8 of the conclusions?

**Response:We have  revised the text and changes or consistency**

MJO is mentioned in the abstract but not in the conclusion

**Response: the MJO is mentioned in the text and also in the discissions on gthe MJO and onset are now shown in the text.**

Please review figure captions. It seems that in Figure 8 the order is not the same as in the figure. What happened to the Bowen ration during the rainy season, is there lack of data?

**Reponse:  capitions have been revised and modified top be consistent with the figures.**
This should be accepted (with some mandatory revision) because it was written by the great José Marengo.

**Response:  Thanks, will work on the revision.**

The comments that I would require to be answered before acceptance are the following:

Page 1, Line 31: "...and also helping the developing Hadley and Walker circulation." Please either correct or elaborate on this phrase. Why is the Hadley circulation developing? Do you mean within the context of the seasonal cycle? And how does Amazon convection participate in the Walker circulation? My understanding is that the Walker circulation is an Indo-Pacific phenomenon.

**Response:  It is in the context of the seasonal cycle and more relevant to the Hadley circulation.  In fact, the reviewer is right, the Walker circulation is a Pacific phenomenon, and what we have in the Indian, African and South America are east-west circulations, not the Walker cell.  We have corrected the text and considered Only East-West circulations.**

P1, L35: Can you support your statement that Atlantic SSTs affect Amazon precipitation with a sentence or two, similar to your description of the Pacific influence? And I believe it would be of value to describe, again briefly, the nature of the teleconnections in both the Atlantic and Pacific. For example, I understand the Pacific teleconnection works by shifting the Walker Circulation to the east, so during El Nino there is more subsidence over the Amazon. Since I have no idea how the Atlantic influences the Amazon, it would be nice to have an idea. Again, nothing new here, just remind the reader what other studies have concluded with a physical explanation. It is nice to have a picture of what is going on.

**Response:  Rainfall variability Amazonia is linked to El Nino, but EL Nino is not the only responsible for rainfall variations in Amazonia, the tropical Atlantic Ocean also plays an important role.  While we had several droughts in Amazonia linked to El Nino, as in 1925, 1983, 1987, 1998 and recently in 2015, some other droughts events have been reported in 1963 and 2005, not related to El Nino but to a warmer topical North Atlantic.  There are several studies that show that and I have listed them in a review paper (Marengo and Espinoza 2016) published in IJOC, and the various studies on the effect of tropical Atlantic in rainfall in Amazonia are lusted in the reference list of that paper.  When the tropical North Atlantic is warmer than the tropical South Atlantic, the intertropical Convergence zone is moved northward leaving less rainfall in the region.  This may happen at the same time with El Nino (1983, 1998) or without an El Nino (2005).  Every drought in Amazonia is different in terms of spatial coverage.**

2-8: How have people's perceptions been changed? Please explain a little better with some detail.

**Response:  In Amazonia, drought is perceived by the population as anomalously low river levels during the peak season May-July, and not much as low rainfall during the peak season en February-April.  Drought is an impact while deficient**

**rainfall is the climatic forcing of this impact. Mau be for some ecological impacts or agriculture drought may be more related to less rainfall during the peak of the rainy season.**

2-10: I think it should be the plural, "show" but it is a complicated sentence.

**Response: Yes, correction will be made.**

2-13: "has"

**Response: Yes, correction will be made.**

2-22: "Variability" sentence does not actually make sense. "Variability" suggests variation, not necessarily long-term change.

**Response: Yes, correction will be made. It is long term variability, without going into climate change time scales.**

2-24 "70s"

**Response: Yes, correction will be made.**

2-27: "While it is important to know how will be" - badly written

**Response: Sorry, correction will be made.**

2-29: "season"

**Response: Yes, correction will be made.**

2-33: Explain what you mean by, "problems in the hydrology of the region."

**Response: This refers to anomalously river levels due to a poor rainy season.**

3-9: This sentence repeats itself: "This may be due to the poor representation of clouds and land surface-atmosphere interactions or due to role of aerosols and other particles, which are still not well represented in models."

**Response: Yes, correction will be made.**

3-14: What do you mean, " Li and Fu (2006) showed that weak and infrequent extratropical cold front penetrations during the transition season also contribute to a delay of the wet season onset?" I presume you mean weaker and less frequent than usual, but if you do, you need to be specific.

**Response: Yes, the reviewer is right, correction will be made.**

4-2: What? "On the regional scale circulation features, during DJF2015 it did not show signals of..."

**Response: Sorry, we do not understand this comment. We did look at line 2 in page 4, and did not find the statement above mentioned on this page.**

4-34: Liebmann and Marengo used gridded rain data.

**Response: Yes, we are ware of that and correction and will make this clear in the text.**

I give up on the writing. Suffice it to say that it is badly written and needs improvement.

**Response: Sorry if the reviewer finds the text badly written. Once the review process is over we will submit to text to a proof reading specialist in the US.**

Please use scaleable, or "vector" graphics. Your rasterized graphics appear fuzzy and thus unprofessional. Figure 2 is not of acceptable quality. In addition to printing it using scaleable graphics, it needs latitudes and longitudes, continental outlines one can see, and perhaps fewer vectors.

**Response: Sorry, we have prepared these figures for the review process only, and we are preparing new and improved figures that will consider all suggestions from the reviewer.**

6-2: horribly written sentence: " On the regional scale circulation features, during DJF2015 it did not show signals of El Niño in the tropical Pacific while the warm surface waters are already present during MAM 2015..."

**Response: Sorry if the reviewer finds the text badly written. We will correct this other unclear statements along the text.**

6-6: From what do you infer reduced northeasterly trades? Is it from the vectors, even though the quantity present by the vectors is the integral to 500 hpa, or is it from assuming flow nearly parallel to surface contours?

**Response: While the anomaly vectors in Figure 2 shows the small wind anomalies suggesting weakened northerly flow, we will include a new figure for the low-level circulation patterns (850 hPa). The 850 wind maps from CPTEC INPE show in fact reduced Northeast trades during January to April 2015.**

6-5: Assuming that you are using the vectors to make the statement, "The low level circulation over the tropical North- Atlantic and Amazon sectors (Figure 2) showed reduction in the Northeast trades....," I disagree (assuming my guess about the map domain is correct). Yes, along the equator (assuming Fig. 2 is centered on the Equator), there are westerly anomalies, but these are away from the coast (looking

at DJF). Along the Atlantic coast and north of the equator, however, the anomalies are nearzero. There are huge positive transport anomalies from the equator into the southern Amazon, which are consistent with above-normal precipitation to the west (south of the equator), as there appears to be anomalous convergence of moisture there (Fig. 2a). So, please explain why this is inaccurate and why your statement is correct.

**We noticed some errors in our explanation and we thank the reviewer for making this visible to us, and we will work on corrections in the text. In fact, we believe that the text above indicated provided by the reviewer better reflects the rainfall-circulation situation, and so we used it in replacement of our previous text.**

6-7: Why are Figures 1 and 2 made from seasonal averages, while Fig. 3 is from monthlies? Would it not be better and certainly more consistent to use seasonal averages in Fig. 3?

**Response: Figures 1 and 2 provide the context of rainfall and circulation detected during austral summer and fall of 2014. Figure 3 is more concentrated on the months where the onset of the rainy season occurs, mainly for January 2015. We consider that having Figure 3 for seasonal time scale it may miss the signal of upward and downward motions linked to development of convection and rainfall along the equatorial region and over Amazonia.**

6-10: The authors may have a valid point, but I believe they should hone in more on Brazil. Nothing is discussed east of the GM, so why not just show the longitudes of South America, plus or minus a bit? And perhaps the shading interval on the anomalous maps should be lowered, as with the present interval it doesn't look like much is going on over South America.

**Response: We chose longitudes beyond South America because we wanted to see the signals of El Nino in 2015 in other regions as well as over Amazonia. We will change the shading interval as the reviewer suggested.**

6-13: " Therefore, interannual variations of the wet season onset in the Amazon appear to be influenced by changes in large scale and regional circulation over the tropical and Pacific sectors."

**Response: We realized that there is a missing word, it should be: Therefore, interannual variations of the wet season onset in the Amazon appear to be influenced by changes in large scale and regional circulation over the tropical _Atlantic_ and Pacific sectors."**

6-21: Instead of, "meaning a rainy season shorter than normal" how about, "meaning a rainy season that was shortened at both ends."

**Response: Thanks, we will do as the reviewer suggested.**

Figure 4 is a little disturbing to me because it does not appear the INMET and UEA records match very well the NOAA records. Looking at the bottom record (Manacapuru), there is no rainfall at all within several days of onset, and it continues to rain for at least a week or so after the NOAA end. I know Manaus is a long record and I assume so is Manacapuru, so why not use the daily station data to do the onset and end calculations? You know that the actual station data is the best record available, so I don't see any reason to use NOAA. I think your point could be made more succinctly and more accurately.

**Response:   We found daily rainfall from INMET data from Manaus from 1961 to 2016, while data from Manacapuru is available from 2008 to 2016.  There are some gaps on the information so we have to make some analyses for data consistency and homogeneity. We will consider re making the figure using rainfall data accumulated in pentads and not n daily data.  If the data is consistent and available for 204-2016, we will re-do Figure 4 and also re calculate the onset and end of the rainy season using Liebmann & Marengo's criterion but applied to the grid box that contains Manacapuru and Manaus, and no longer using the NOAA data for this.**

6-25: "which are not common for the wet season." This cannot be stated without any sort of justification, such as a reference.

**We will add some references to support this statement.**

Conclusions: Please make sure your conclusions match your discussion in the Results section. For example, you discussed the change in moisture transport (which I disagreed with), but this discussion did not make it into the conclusions.

**Response:  Thanks, we will work on that.**

Good luck - Brant Liebmann

**Response:  Thanks greater Brant, really appreciate your suggestions and input that will improve the paper.**

**General response:  We are redoing some of the figures, and as soon we get the comments from all reviewers we will incorporate them on the text and produce a new version, that will be sent to a professional proof reader for editing and text correction, and we will submit that revised version to ACP.**

**Response to reviewer S. Ferraz**

This should be accepted (with little revision) because it is a very interesting theme. This research has great impact, since the determination of the dry and rainy season of the Amazon is important not only locally but for the distribution of humidity in the rest of the South American continent. Studies with high frequency data are highly desirable as they describe in detail the characteristics of the region. In this context, this work is able to encompass the local characteristics of the region, with data from

the experiments Go Amazon and Rain Project and the large scale characteristics due to the ENSO phenomenon. In this study the authors suggest that the dry season experienced has impacts of the En Nino and the beginning of the rainy season may be related to the MJO.

- How does the relationship between MJO and ENSO occur? A brief explanation of this relationship would contribute to a better understanding of the article. - It is known that different phases of the MJO influence the precipitation in South America in different seasons. What is the climatological influence of MJO in this?

**Response:**

**Broadly speaking, the influence of the MJO on precipitation over the tropics occurs by eastward propagation of Rossby wave trains from the tropical Pacific Ocean (Muza et al. 2009). Previous observational and modeling studies generally indicated that MJO and ENSO has a decadal variation and seasonal dependence (Tang and Yu, 2008; Hendon et al. 2007), however, them has not been well identified due to nonlinear in nature. These studies also show significantly lagged correlations between MJO and ENSO indices.**

**Despite this, recently, Shimizu et al. (2016) examined the regional relationship between ENSO and MJO phases on climatological patterns of precipitation over South America. The results indicated that combined responses showed that precipitation is strongly influenced by the MJO phases rather than by ENSO conditions, especially during the austral summer. Then, our results corroborate with Shimizu et al. (2016) who observed highest percentages of days with active MJO occurred during El Niño and neutral years and an increase of precipitation.**

Espinoza 2015, and references quoted therein). Indeed, previous studies have documented that warm conditions in the

Pete 8/5/17 18:13

Pete 4/5/17 22:55

Pete 8/5/17 18:14

Pete 8/5/17 18:14

Pete 8/5/17 18:15

Pete 8/5/17 18:16

Pete 8/5/17 22:54
**Comment [1]:** better to say zonal and vertical components of the circulation ?(see comment 5)

Pete 4/5/17 23:00

Pete 4/5/17 23:00

Pete 4/5/17 23:00

User 2/5/17 13:31

equatorial Pacific (e.g. El Niño events) produce a rainfall deficit in Amazonia, which can originate extreme drought periods, as observed in 1926, 1983, 1997-1998 and 2010 (Williams et al. 2005; Marengo et al. 2008, 2011; Espinoza et al. 2011; Marengo and Espinoza 2015). In the early 21[st] century, large-scale extreme seasonal events, such as extreme droughts in 2005, 2010 and 2015 and floods (2009, 2012, 2014) have affected the Amazon region. Rainfall anomalies were the consequence of circulation changes forced by anomalous warming or cooling of the tropical Pacific and/or tropical north or south Atlantic Oceans as documented by Marengo and Espinoza 2015 and references quoted therein).

In sum, while we had several droughts in Amazonia linked to El Nino, as in 1925, 1983, 1987, 1998 and recently in 2015-16, some other drought events have been reported in 1963 and 2005, not related to El Nino but to a warmer topical North Atlantic. When the tropical North Atlantic is warmer than the tropical South Atlantic, the intertropical convergence zone is displaced northward leaving less rainfall in the region. This may happen with an El Nino (1983, 1998) or without an El Nino (2005). Every drought in Amazonia is different in terms of spatial coverage.

[revised manuscript text omitted]

Lincoln Muniz 19/4/17 11:31

Unknown

Lincoln Muniz 19/4/17 11:39

Lincoln Muniz 19/4/17 11:39

Lincoln Muniz 19/4/17 11:39

Lincoln Muniz 19/4/17 11:40

Lincoln Muniz 19/4/17 11:40

Lincoln Muniz 19/4/17 11:44

[Figure]

**Figure 5.** Hovmoller diagram of 30-60-day filtered OLR anomalies (W m$^{-2}$) along the Equator (5ºN-5ºS) during the period of June 2014 to May 2015.

[Figure]

**Figure 6. Daily variation of relative humidity (%), 850 hPa air temperature (°C) and Bowen ratio at the T3 Manacapuru site, from January 2014 to July 2015. Blue bar indicates the onset and end of the rainy season for 2014-15.  No Bowen ratio data were available during some days between January 2014 and July 2015.**

Pete 7/5/17 17:50

[Figure]

Figure 7. Daily variation of CAPE and CIN (J kg$^{-1}$) from January 1 to February 25, 2015 (around the onset of the rainy season) and from April 11 to June 30 (around the demise of the rainy season) at Manaus. Pentads of the onset

10   and demise are identified with blue vertical bars.

User 2/5/17 15:17

Pete 7/5/17 17:51
Pete 7/5/17 17:51

[Figure]

Figure 8. Diurnal cycle of the height of boundary (m) layer (average from January 1 up to June 30, 2015) (a); from January 2015 (b) and June 2015 (c).

---

## Author Response (AR3)

**Co-Editor Decision: Publish subject to technical corrections** (23 May 2017)
by Maria Assuncao Silva Dias

Comments to the Author:

In the response to the reviewer comment 6-13 the authors mention that the word `Atlantic' should be included. In the new version of the manuscript, page 6 line 24, the word Atlantic is still missing.

**Response:  we have corrected those mistakes**

Page 8 line 15 - the sentence seems broken, please correct.

**Response:  we have corrected those mistakes**

Figure 8 - need to label each part of the figure with (a), (b) and (c).

**Response:  we have corrected the figure**